# The bat influenza A virus subtype H18N11 induces nanoscale MHCII clustering upon host cell attachment

Maria Kaukab Osman [1,2,3,4,11], Jonathan Robert [1,2,3,4,11], Lukas Broich [5], Dennis Frank[6], Robert Grosse [6,7], Martin Schwemmle [1,2], Antoni G. Wrobel [8,9], Kevin Ciminski [1,2], Christian Sieben [5,10] & Peter Reuther [1,2]

Prior to the discovery of bat influenza A virus (IAV) subtypes H17N10 and H18N11, all IAVs were thought to bind sialic acid residues via hemagglutinin (HA) to mediate attachment and subsequent viral entry. However, H17 and H18 engage a proteinaceous receptor: the major histocompatibility complex class II (MHCII). The mechanistic details of this hitherto unknown protein-mediated entry are not understood. Given that conventional IAVs rely on multivalent binding to sialylated glycans, we hypothesized that bat HA similarly interacts with multiple MHCII molecules. Using photoactivated localization microscopy (PALM) on fixed and live cells, we demonstrate that bat IAV particles attach to pre-existing MHCII clusters and induce a further increase in cluster size upon binding. To measure the impact of viral attachment on the dynamics of MHCII, we employ an "inverse attachment" approach, immobilizing viral particles on coverslips before seeding live MHCII-expressing cells on top. Single-molecule tracking reveals that the mobility of MHCII is indeed slowed down in viral proximity leading to a local enrichment of MHCII molecules beneath the viral particle. These findings suggest that viral attachment induces MHCII clustering, a process similar to the MHCII dynamics observed during the formation of an immunological synapse.

Influenza A viruses (IAV) have a major impact on global health causing annual epidemics and sporadic pandemics. Until 2012, aquatic birds were believed to represent the sole reservoir of IAV maintaining all previously known hemagglutinin (HA) (H1-16) and neuraminidase (NA) (N1-9) subtypes[1–3]. This notion changed with the discovery of two novel IAV subtypes, designated H17N10 and H18N11, in bats from Central and South America[4–6]. While these bat IAVs structurally resemble conventional IAVs of avian origin, their HA and NA surface glycoproteins are functionally distinct. In contrast to conventional HAs, the bat IAV HAs (H17 and H18) do not bind sialic acids for cell entry and the correspondent NAs (N10 and N11) lack sialidase activity[5,7–9]. Previous studies using recombinant vesicular stomatitis virus (VSV) expressing either

[1]Institute of Virology, Medical Center—University of Freiburg, Freiburg, Germany. [2]Faculty of Medicine, University of Freiburg, Freiburg, Germany. [3]Spemann Graduate School of Biology and Medicine, University of Freiburg, Freiburg, Germany. [4]Faculty of Biology, University of Freiburg, Freiburg, Germany. [5]Nanoscale Infection Biology Group, Department of Cell Biology, Helmholtz Centre for Infection Research, Braunschweig, Germany. [6]Institute of Experimental and Clinical Pharmacology and Toxicology, Medical Faculty, University of Freiburg, Freiburg, Germany. [7]Centre for Integrative Biological Signaling Studies—CIBSS, Freiburg, Germany. [8]Structural Biology of Disease Processes Laboratory, The Francis Crick Institute, London, UK. [9]Department of Biochemistry, University of Oxford, Oxford, UK. [10]Institute of Genetics, Technische Universität Braunschweig, Braunschweig, Germany. [11]These authors contributed equally: Maria Kaukab Osman, Jonathan Robert. ✉e-mail: christian.sieben@helmholtz-hzi.de; peter.reuther@uniklinik-freiburg.de

H18 or N11, have shown that H18 but not N11 is sufficient to mediate cell entry and allow viral spread[10]. Consistent with these observations, we demonstrated that a mutant H18N11 lacking the N11 ectodomain (designated rP11) not only replicates efficiently in cell culture but also in its natural host the Jamaican fruit bat (*Artibeus jamaicensis*)[11]. Even though N11 is still required for efficient transmission among bats, H18 thus seems to be the key determinant of effective infection at the cellular level. However, the receptor(s) involved in H18-mediated entry remained unknown until recently.

We and others showed that bat IAVs rely on a proteinaceous receptor for cell entry: the major histocompatibility complex class II (MHCII)[12,13]. MHCII is a heterodimeric transmembrane protein complex consisting of an α and β chain, each comprising two extracellular domains: α1, α2 and β1, β2[14]. MHCII is mainly expressed on professional antigen-presenting cells (APC) such as macrophages, dendritic cells and B cells[15]. Here, MHCII has an essential role in adaptive immunity by presenting peptides from endocytic compartments to $CD4^+$ T cells. Interestingly, MHCII from various vertebrate species including the human leukocyte antigen DR (HLA-DR), support H17 and H18-mediated infection and the highly conserved amino acid residues within the α2 domain of MHCII are required for cell entry[12,16]. However, the initial steps of bat IAV infection, including receptor engagement, endocytosis and endosomal release, remain elusive. In-depth analysis of receptor engagement by classical biochemical approaches has been unsuccessful likely due to a low affinity between bat HA and MHCII, a feature reminiscent of the HAs of conventional IAVs, which bind sialic acids very weakly[17–19]. So far, an MHCII-H18 interaction could only be confirmed by chemical crosslinking on the cell surface[12].

Conventional IAV particles interact with sialylated cell surface glycoproteins[20], which are organized in submicrometer nanoclusters[21,22]. These clusters represent multivalent virus binding platforms that provide the avidity necessary for attachment of multiple low-affinity HAs and subsequent endocytosis of the viral particle. Based on a previous observation that MHCII is enriched in membrane clusters of APCs, we hypothesized that these MHCII clusters also serve as multivalent attachment sites[23–25]. As our recent in silico model suggests that one H18 homotrimer can bind three MHCIIs, we speculate that upon attachment of bat IAV particles, additional MHCII complexes are recruited[16].

Super-resolution microscopy represents a powerful tool to study interactions of IAV proteins and associated host factors[21,26–29]. Here, we use photoactivated localization microscopy (PALM) to visualize the nanoscale organization of MHCII and the interaction dynamics of bat IAV and MHCII in live cells at the single-molecule level[30,31]. We show that individual bat IAV particles interact with clusters of MHCII, resulting in decreased mobility of the viral particle at the cell surface. Using an "inverse attachment" approach, which allows to study virus-receptor interactions on live cells by PALM, we show that additional MHCII molecules are trapped at the virus-cell interface. This results in increased MHCII cluster size, suggesting that viral particles induce nanoscale MHCII clustering upon host cell attachment.

## Results

### Cells expressing MHCII fused to the photoconvertible fluorescent protein mEos3.2 are highly susceptible to infection with the bat IAV H18N11

To visualize the dynamics of MHCII upon attachment of H18N11, we transduced non-permissive MDCK-II cells with a DNA cassette from which the wildtype (wt) alpha (α) and beta (β) chain of the human MHCII HLA-DR15 are expressed (Fig. 1a). As we have shown recently that covalent fusion of a model peptide to MHCII enhances the susceptibility to bat IAV infection, the antigenic $HA_{307-319}$ peptide was linked to the extracellular N-terminus of the beta chain[16]. To allow visualization of MHCII at the single-molecule level in live cells by PALM, we fused the photoconvertible fluorescent protein mEos3.2 to the

intracellular C-terminus of the β chain ($MHCII_{mEos}$)[32]. In the native state, mEos3.2 emits green fluorescence, but can be irreversibly converted by UV-illumination at 405 nm. Photo-converted mEos3.2 emits red fluorescence when excited with a 561 nm laser before it eventually photobleaches (Fig. 1b, Fig. S1a). In PALM, this photoconversion mechanism is used to localize individual mEos3.2 molecules over time allowing reconstructing of the organization and dynamics of single MHCII molecules on the cell surface. We also generated a mutant MHCII ($MHCII_{mEosmut}$) with 11 amino acid substitutions in the α2 subunit derived from the non-classical MHCII, HLA-DM (Fig. 1a), which we recently demonstrated to not support bat IAV infection[16]. Both $MHCII_{mEos}$ and $MHCII_{mEosmut}$ were efficiently expressed at the cell surface of transduced MDCK-II cells (>99%) (Fig. 1c, Fig. S2a). Furthermore, both wt and mutant MHCII, here fused to the red-fluorescent TagRFP ($MHCII_{TagRFP}$ and $MHCII_{TagRFPmut}$), activated T cells in a haplotype-specific manner confirming the overall structural integrity of the proteins (Fig. 1d, Figs. S1b–d, S2b).

Next, we compared the ability of each of the two MHCII constructs to support bat IAV infection. While expression of $MHCII_{mEos}$ rendered cells susceptible to infection with the H18N11 variant rP11, $MHCII_{mEosmut}$ did not (Fig. 1e, Fig. S1e, Fig. S2c). Using H18N11 virus-like particles (VLPs) harboring an M1-β-lactamase (M1-Bla) fusion protein, we further probed at which stage the infection is blocked[12]. We detected no cytoplasmic Bla activity in the $MHCII_{mut}$-expressing cells proving that no viral release from the endosome occurred (Fig. 1f, Fig. S2d). In addition, H18-mediated cell fusion was only observed in presence of the wt but not the mutant form, demonstrating that H18 is unable use mutant MHCII for membrane fusion (Fig. 1g, Fig. S1f).

To test whether the incompatibility of $MHCII_{mEosmut}$ with H18N11 infection is due to an insufficient interaction on the cell surface, we produced soluble ectodomains of wt MHCII (sMHCII) and mutant MHCII ($sMHCII_{mut}$) and tested their ability to neutralize H18N11 (Fig. 2a–c). sMHCII and $sMHCII_{mut}$ were purified from the supernatant of transfected Expi293F cells and incubated with H18N11 prior to infection of MDCK-II cells stably expressing $MHCII_{mEos}$. Under these conditions, sMHCII efficiently neutralized H18N11 particles in a dose-dependent manner, as determined by counting the number of infectious viruses (focus forming units) left after treatment (Fig. 2d). No virus neutralization was observed for $sMHCII_{mut}$.

In summary, our fluorescently-labeled wt MHCII supports bat IAV infection and is therefore suitable for further functional studies using high-resolution microscopy. Cell surface-expressed $MHCII_{mEosmut}$ will serve as an adequate negative control, since it does not support viral infection, but is still able to activate T cells.

### H18N11 particles bind to MHCII clusters on the cell surface

Since conventional IAVs overcome their poor affinity to sialic acid by multivalent binding to clusters of sialylated glycans on the cell surface[21,33], we hypothesized that H18N11 may preferentially bind to preformed clusters of MHCII. To visualize the molecular organization of MHCII on the cell surface, MDCK-II cells expressing either $MHCII_{mEos}$ or $MHCII_{mEosmut}$ were fixed and imaged with PALM (Fig. 3a). Analysis of nanoclustering revealed clusters on the cell surface with a comparable median radius of gyration (Rg) of 33 nm for wt MHCII and 40 nm for mutant MHCII (Fig. 3b, c, Fig. S3a–c, Fig. S4a, Fig. S5a–g). This range of cluster sizes is in agreement with the previously observed organization of MHCII expressed endogenously on professional APCs[34]. While we determined consistent resolution estimates of our datasets (Fig. S5b, d), the cluster density as well as the number of MHCII molecules per cluster was lower for the mutant MHCII, the latter suggesting that the mutation also has an impact on the intermolecular affinity within a cluster (Fig. S3d–e).

Next, we set out to determine the size of MHCII clusters interacting with H18N11 viral particles. Cells were exposed to the H18N11 variant rP11[11] on ice to allow attachment to the cell surface but without

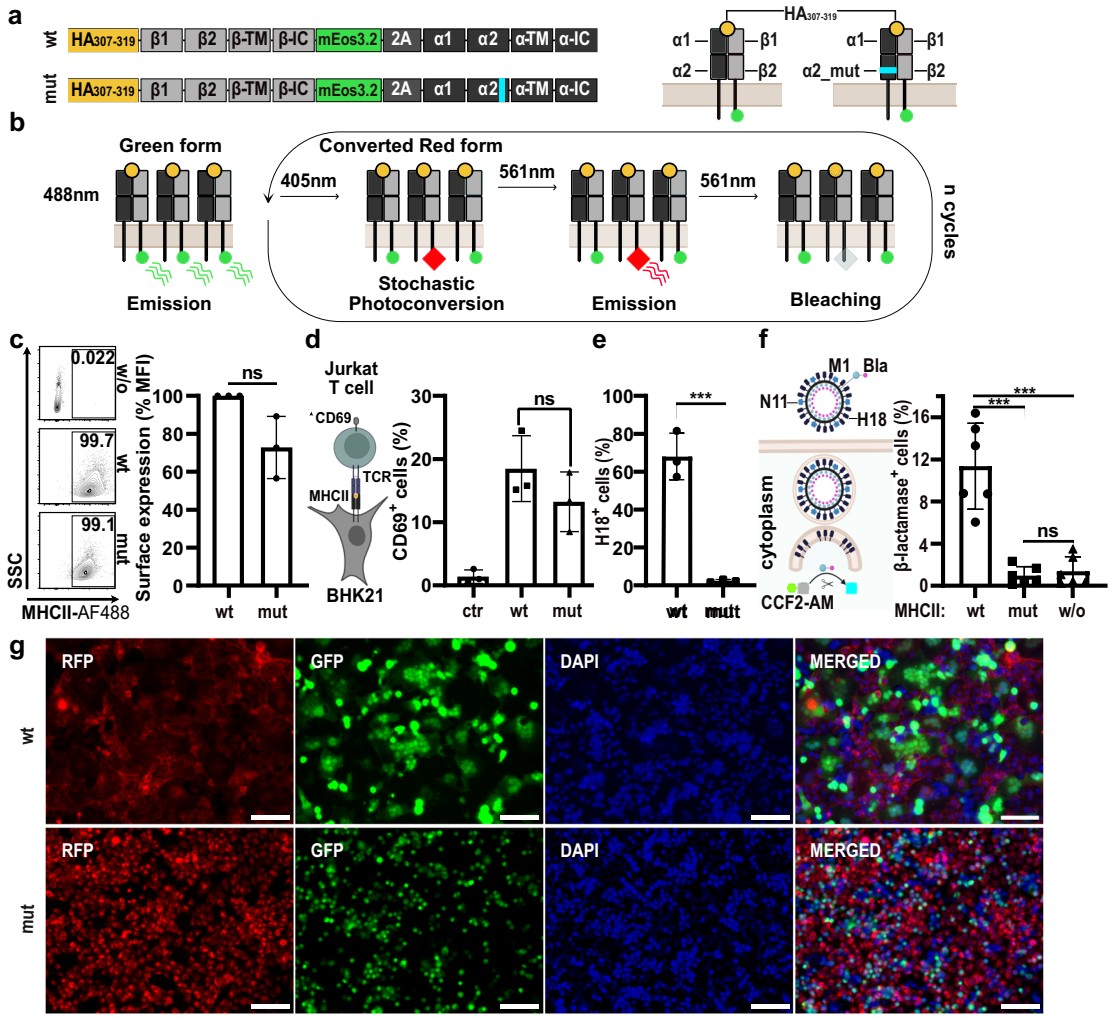

**Fig. 1 | Fluorescently-labeled MHCII mediates entry of bat IAV. a** Illustration of the MHCII (HLA-DR) constructs: $MHCII_{mEos}$ and $MHCII_{mEosmut}$. The N-terminus of the beta chain is fused to the $HA_{307-319}$ peptide and the C-terminus is linked to the mEos3.2 protein. α2 of $MHCII_{mEosmut}$ comprises a substitution of 11 amino acids (cyan). **b** Photoconversion of mEos3.2. Upon 405 nm light exposure, mEos3.2 converts stochastically. Converted mEos3.2 emits red light upon 561 nm excitation and then photobleaches. **c** Analysis of the surface expression of $MHCII_{mEos}$ (wt) and $MHCII_{mEosmut}$ (mut) in transduced MDCK-II cells. Numbers within plots indicate the percentages of $MHCII^+$ cells. Bar graph depicts the median fluorescent intensity normalized to wildtype from $n = 3$ independent experiments. **d** Cartoon illustrating the T cell activation assay. BHK21 cells transiently expressing $MHCII_{TagRFP}$ (Haplotype: HLA-DR1), are co-cultured with CH7C17Jurkat T cells, which recognize $HA_{307-319}$ in combination with HLA-DR1. Activated T cells express CD69. Bar graph shows the number of CD69 positive T cells from $n = 3$ independent experiments. $MHCII_{TagRFP}$ of the haplotype HLA-DR15 served as control (ctr.). **e** Flow cytometric analysis of the infection rate of MDCK-II cells stably expressing $MHCII_{mEos}$ (wt) or $MHCII_{mEosmut}$ (mut) at 24 h post-infection with H18N11 at an MOI of 5 from $n = 3$ independent experiments. **f** Quantification of viral entry using H18N11 VLPs harboring the viral matrixprotein (M1) fused to a β-lactamase (M1-Bla). Bla activity in the cytoplasm was determined with the FRET-based reporter CCF2-AM. Bar graph depicts the percentage of β-lactamase positive cells expressing $MHCII_{TagRFP}$ (wt), $MHCII_{TagRFPmut}$ (mut) or no MHCII (w/o). $n = 6$ independent experiments. **g** pH-induced polykaryon formation of HEK293T cells expressing H18 and GFP with MDCK-II cells stably expressing $MHCII_{TagRFP}$ (wt) or $MHCII_{TagRFPmut}$ (mut). Representative images from $n = 3$ independent experiments. Scale bar 100 μm. **c–f** Data are mean ± SD. Paired two-tailed Student's t-test was applied for (**c**) and unpaired two-tailed Student's t test was used in (**d–f**). **e** ***$p = 0.0066$, **f** ***$p = 0.0039$ for wt vs. mut, ***$p = 0.0331$ for wt vs. w/o, ns not significant. **d, f** Schemes were created with BioRender.com (https://BioRender.com/c35p533). **c–f** Source data are provided as a Source Data file.

subsequent internalization of viral particles. PALM revealed $MHCII_{mEos}$ co-localizing with single viral particles after fixation and anti-H18 immunolabeling. We found that clusters underneath viral particles are larger with a median Rg of 48 nm, compared to the median cluster size of Rg = 29 nm at virus-free surfaces (Fig. 3d, e). In sharp contrast, the median cluster size of $MHCII_{mEosmut}$ was found to be 37 nm under viral particles (Fig. 3f, g). Taken together, viral particles bind to $MHCII_{mEos}$ with increased cluster size on ice, suggesting a direct interaction of H18 viruses with pre-existing MHCII clusters. To exclude the possibility that our analysis was confounded by large H18-containing aggregates and/or exosomes, we thoroughly evaluated our virus preparation. Most of the particles stained positive for H18 by immunolabeling and

also comprised viral nucleoprotein (NP) (Supplementary movie 5, Fig. S6a). Furthermore, transmission electron microscopy analysis revealed almost exclusively the presence of viral particles resembling pleomorphic H18N11 virions in size and shape as shown previously[10] (Fig. S6c).

## Interaction between H18 viruses and MHCII receptors decreases the mobility of both, the viral particles and MHCII

For conventional IAV, we have recently shown that viral particles become transiently confined to the host cell surface when they encounter a region of high receptor density[21]. Similarly, we hypothesized for H18N11 particles that binding of MHCII may decrease both,

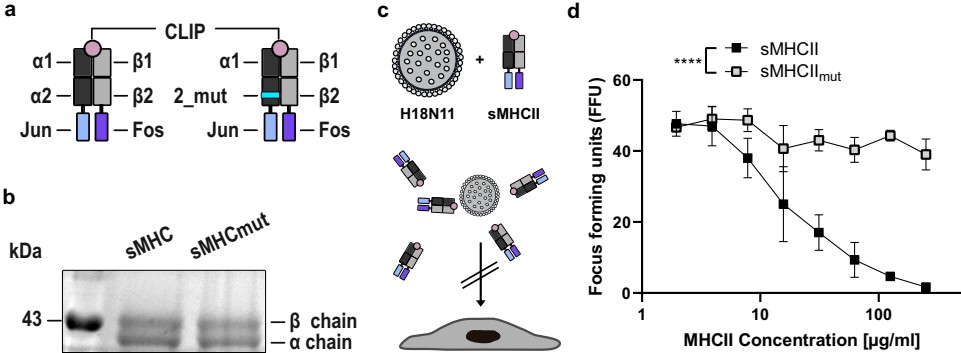

**Fig. 2 | Soluble, wildtype MHCII efficiently neutralizes H18N11 viral particle.** **a** Illustration of soluble MHCII (sMHCII) and mutant (sMHCII_mut). The C-termini of the α and the β chain were fused to the zipper motifs Jun and Fos, respectively. The N-terminus of the β-chain is fused to the CLIP peptide. The 11 amino acid substitution within the sMHCII_mut α-chain is highlighted in cyan. **b** Coomassie staining of the purified sMHCII and sMHCII_mut. **c** Schematic illustration of the H18N11:MHCII competition assay. H18N11 viral particles are incubated in presence of soluble MHCII prior to infection. **d** Comparison of the ability of the indicated concentrations of sMHC and sMHC_mut to neutralize 50 focus-forming units of H18N11. Data are mean ± SD of n = 4 independent experiments. Two-way ANOVA was used for statistical analysis, **** $p < 0.0001$. **a**, **c** from $n = 3$ independent experiments. **b**, **c** Source data are provided as a Source Data file.

the viral mobility and the local receptor diffusion leading to larger clusters as observed in fixed cells (Fig. 3d, e). To visualize the dynamic movement of H18N11 on the cell surface, particles were fluorescently labeled and viral trajectories were recorded by spinning-disk confocal microscopy (Fig. 4a). For this purpose, MHCII-expressing cells were incubated with DiO-labeled H18 particles at 4 °C to allow virus-cell binding while preventing uptake. Subsequently, cells were transferred to 37 °C and trajectories were recorded over 20 min. This analysis revealed short viral trajectories on cells expressing MHCII_TagRFP, whereas trajectories were significantly longer on MHCII_TagRFPmut cells (Fig. 4b). This suggests a confinement of the H18N11 particle on the cell surface in the presence of wildtype MHCII. Longer viral trajectories observed on the surface of cells expressing mutant MHCII may likely result from a reduced affinity to H18 (Fig. 2d) and/or a lower cluster density (Fig. S3d).

We further speculated that the H18-MHCII interaction does not only reduce the mobility of the viral particle, but that viral attachment also spatially confines MHCII. Tracking individual MHCII molecules relative to the viral particle by live-cell PALM at physiological temperature, however, is challenging, as viral particles are rapidly internalized. To circumvent this problem, we established an experimental setup designated "inverse attachment"[35] (Fig. 4c). Here, fluorescently-labeled viral particles were covalently immobilized on reactive epoxy-modified glass slides. MDCK-II cells expressing MHCII_mEos or MHCII_mEosmut were seeded on top of the immobilized viral particles and we performed live-cell single-particle tracking (spt)PALM. Of note, the majority of the fluorescently-labeled particles was also H18 positive and the resolution estimates for MHCII_mEos and MHCII_mEosmut were consistent (Supplementary movie 6, Fig. S6b, Fig. S7a, b). The inverse attachment approach revealed that MHCII_mEos molecules diffuse freely in virus-free surface areas, while their tracks were more condensed above the viral particles (Fig. 4d, Fig. S4b, e). Notably, we did not observe this degree of immobilization of MHCII with conventional IAV A/PR/8/34 (H1N1) suggesting a virus-specific interaction (Fig. 4e). To better quantify the local confinement, we calculated the mean squared displacement (MSD) of all measured MHCII trajectories. An MSD vs. lag time plot can be fitted to a power-law distribution (see "methods") where the power-law exponent (α) can be used as a measure for local confinement[36]. While α of 1 indicates free diffusion, smaller α values indicate restricted lateral mobility. As shown in Fig. 4f, the contact with an H18N11 particle resulted in local confinement of MHCII_mEos (median α = 0.64), whereas the mobility of MHCII_mEos in proximity of an H1N1 particle was almost not restricted (median α = 0.8). The local

confinement of MHCII_mEos in contact with H18N11 particle was significantly higher than that of MHCII_mEosmut. Yet, for MHCII_mEosmut, some degree of immobilization above the viral particle was observed compared to uninfected cells suggesting a residual affinity of the mutant MHCII to H18 (Fig. 4f). Virus-induced receptor confinement as observed here should lead to local enrichment of MHCII proteins in proximity of the virus. Indeed, visualization of the MHCII dynamics at the virus-binding site by rendering our sptPALM data in time-binned, time-lapse super-resolution movies revealed that some viral particles were already associated with MHCII clusters at the beginning of the acquisition, but new clusters were formed later during the acquisition period at the virus-binding site (Supplementary movies 1–4, Fig. S4k–n). To gain further insights into the dynamics of MHCII upon viral attachment, we simulated a patch of the plasma membrane with diffusing MHCII receptors. We then added circular regions (i.e., bound viruses) where the receptor diffusion was reduced due to virus-receptor interaction (Fig. 4g). The corresponding diffusion coefficients (D_MHCII and D_MHCII+H18) were taken from the power-law exponent analysis described above (see "methods"). We then simulated 4000 time steps (interval 30 ms) and counted the number of receptors in viral proximity. For the H18-mediated diffusion slowdown measured for MHCII, we indeed found a significant local enrichment as compared to freely diffusing receptors (Fig. 4g).

In summary, we provide evidence that the interaction between H18 and MHCII results in decreased viral mobility on the host cell surface and local confinement of individual receptors in viral proximity. The H18-MHCII interaction slows down MHCII diffusion which leads to a characteristic local enrichment leading to larger MHCII clusters beneath the viral particles (as shown in Fig. 3d, e).

## Discussion

The discovery of influenza A viruses in New World bats changed the paradigm that entry of IAV is strictly glycan-dependent. While MHCII was identified as the primary receptor, detailed insights into the entry of bat IAV have been missing. Here, we reveal that viral attachment to the host cell surface via MHCII is a highly dynamic process. We show that H18N11 associates with large MHCII clusters resulting in a local confinement of the viral particle. This is consistent with observations from conventional IAV, where multivalent receptor-ligand interactions are necessary to decrease the overall dissociation rate[21,22]. In contrast to conventional IAVs, bat IAVs, do not attach to nanodomains composed of diverse sialylated glycans, but rely on a single proteinaceous receptor. Consistently, the mutant MHCII used in this study, which

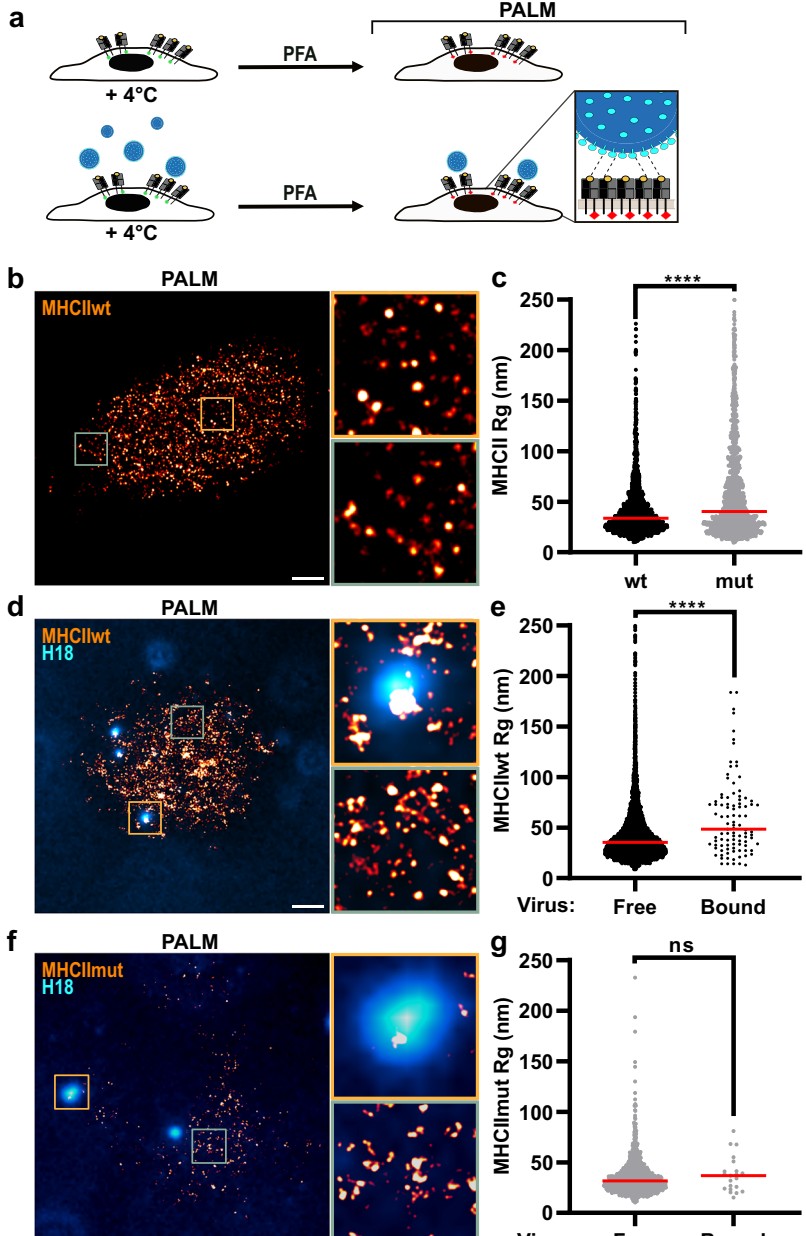

**Fig. 3 | Viral particles are associated with large clusters of wt MHCII but not mutant MHCII. a** Schematic representation of the analysis of the MHCII-distribution at the surface of fixed MDCK-II cells stably expressing mEos-fused MHCII by Photoactivated Localization Microscopy (PALM) in the presence or absence of H18N11 particles. **b** MHCII-distribution on the surface of noninfected MDCK-II cells stably expressing MHCII$_{mEos}$ (wt) by PALM. Representative images from $n = 3$ independent experiments. **c** Comparison of MHCII cluster sizes at the surface of non-infected MDCK-II cells stably expressing MHCII$_{mEos}$ (wt) and MHCII$_{mEosmut}$ (mut) from $n = 3$ independent experiments. Per experiment, we imaged 10–20 cells, each containing 1–5 viruses. **d** MHCII-distribution at the surface of MDCK-II cells stably expressing MHCII$_{mEos}$ (wt) in the presence of H18N11 viral particles determined by PALM. Viral particles were visualized by staining for H18

(cyan). Representative images from $n = 3$ independent experiments. **e** Comparison of MHCII cluster sizes at virus-free and virus-bound surfaces of MDCK-II cells stably expressing MHCII$_{mEos}$ (wt) in the presence of H18N11 viral particles. Data was pooled from at least two independent experiments. **f** MHCII-distribution at the surface of MDCK-II cells stably expressing MHCII$_{mEosmut}$ (mut) in the presence of H18N11 viral particles. Representative images from $n = 2$ independent experiments. **g** Comparison of MHCII cluster sizes at virus-free and virus-bound surfaces of MDCK-II cells stably expressing MHCII$_{mEosmut}$ (mut) in the presence of H18N11 viral particles. Data was pooled from at least two independent experiments. Unpaired, two-tailed Student's t-test was used for comparison. ****$p < 0.0001$, ns not significant. **a** Schemes were created with BioRender.com (https://BioRender.com/c35p533). **c, e, g** Source data are provided as a Source Data file.

carries mutations in the proposed H18 binding site, does not support viral entry. Using the "inverse attachment" approach[35], we can show that the interaction of the bat IAV particle with MHCII also results in a reduced mobility of MHCII. Interestingly, it was demonstrated that cross-linking of MHCII by natural ligands or MHCII-specific antibodies induces downstream signaling, which promotes clathrin-dependent endocytosis[37–40]. It is therefore tempting to speculate that attachment

of H18N11 similarly triggers intracellular signaling resulting in the uptake of MHCII:H18N11 complexes. Likewise, outside-in signaling by receptor tyrosine kinases such as EGFR is also required for the efficient uptake of conventional IAVs[41]. Thus, despite of the fundamentally different nature of the cellular receptor, the early events of the bat IAV entry seem to resemble that of conventional IAVs. Of note, local confinement of receptor molecules triggered by viral attachment was

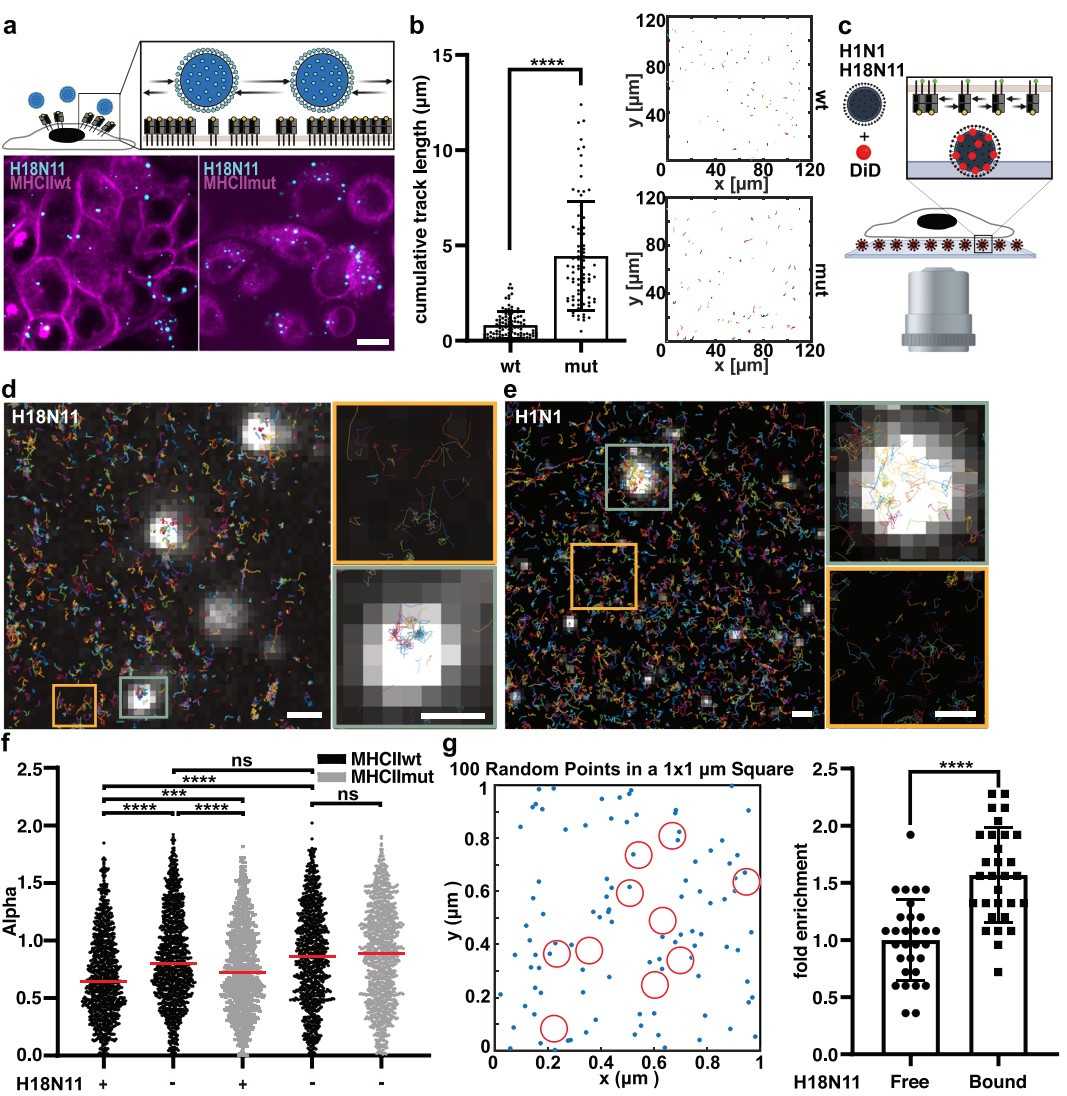

**Fig. 4 | MHCII is enriched underneath H18N11 viral particle upon attachment.** **a** Viral particle tracking on the cell surface. Fluorescently-labeled viral particles were added to cells expressing MHCII$_{TagRFP}$ (wt) or MHCII$_{TagRFPmut}$ (mut). Scale bar 10 µm **b** Representative trajectories of viral particles on cells expressing MHCII$_{TagRFP}$ or MHCII$_{TagRFPmut}$ (right) and comparison of the cumulative track length from one experiment (left). wt: n = 97, mut: n = 79 **c** Inverse attachment assay. Labeled viral particles were immobilized and cells stably expressing MHCII$_{mEos}$ were seeded on top. Tracking of MHCII$_{mEos}$ molecules on top of H18N11 (**d**) or H1N1 (**e**) particles (white). Representative figures from n = 2 experiments. Scale bars: 1 µm (overviews) and 500 nm (zoom-ins). **f** Power-law exponent α of MHCII molecules on the surface of cells exposed to the indicated viruses at virus-bound areas (n = 2 experiments). Mock-infected cells served as control. wt+H18N11: n = 1027, wt+H1N1: n = 1042, mut+H18N11: n = 1017, wt + mock: n = 1021, mut +

mock: n = 1025. **g** To calculate if a slowdown of receptor proteins at the virus binding site leads to local protein enrichment, we simulated the 2D random diffusion of MHCII using our measured diffusion coefficient D$_{MHCII}$ = 0.032 µm²/s. We added circular regions (red, r = 50 nm) to the simulation, where proteins were slowed down according to our PALM measurements to D$_{MHCII+H18}$ = 0.015 µm²/s. We simulated 4000 time steps with 30 ms interval. We examined the number of receptors associated with circular regions (i.e., viruses). Each data point represents one simulation (n = 30). **b, f, g** Unpaired two-tailed Student's t-test was used for pairwise comparison. **b** ****p < 0.0001 **f** ****p < 0.0001 for wt + H18N11 vs. wt + H1N1, ***p = 0.0002 for wt + H18N11 vs. mut + H18N11, ****p < 0.0001 for wt + H18N11 vs. wt w/o, ****p < 0.0001 for wt + H18N11 vs. wt w/o, ns: not significant. **g** ****p < 0.0001. **a, c** Schemes were created with BioRender.com (https://BioRender.com/c35p533). **b, f, g** Source data are provided as a Source Data file.

shown to occur prior to entry of numerous other viruses across diverse families[42–44]. The "inverse attachment" approach described here can be easily adapted to other virus-receptor interactions and could therefore prove to be a valuable tool for investigating the very early and critical events of viral entry.

Subsequent to endocytosis, the IAV envelope has to fuse with the endosomal membrane[45]. Performing cell fusion assays, we could prove that viral-host membrane fusion is also dependent on the interaction between H18 and MHCII[12,16]. Accordingly, the mutant MHCII, which cannot be enriched under the viral particle on the cell surface does not promote membrane fusion. Based on this finding, we hypothesize that

for endosomal escape, viral particles also require a multivalent interaction with MHCII inside the endosome.

The selective pressure that drove viral evolution towards the use of a proteinaceous receptor instead of sialic acid is still unclear, but illustrates that IAVs have the potential to switch their receptor specificity with unforeseen consequences such as a complete change in cellular tropism[46,47]. Of note, a novel IAV subtype (H19), which was identified recently in an avian reservoir, was shown to also depend on MHCII for cell entry[48]. This demonstrates that MHCII-binding is not a unique feature of bat-borne IAV and might be evolutionarily older than anticipated.

In this study, we shed light on the initial steps of the bat IAV H18N11 replication cycle. We show that viral attachment and subsequent uptake depends on a dynamic interaction between H18 on the viral particle and MHCII on the host cell surface. This interaction results in a confinement of both the virus and MHCII, which seems to be critical for the induction of an intracellular signaling cascade that promotes endocytosis.

## Methods

### Cell lines

Human embryonic kidney HEK293T cells were obtained from the American Type Culture Collection (ATCC;CRL3216). Baby Hamster Kidney Fibroblasts (BHK-21) were obtained from the German Cell Culture Collection (DSZM). MDCK-II cells stably expressing MHCII were generated by lentiviral transduction as described[12]. Two days post transduction cells were selected with 2.5 µg/ml puromycin. Puromycin-resistant cells were single-sorted to obtain clonal cell lines. All adherent cells were cultured in Dulbecco's Modified Eagle's Medium (DMEM; Gibco, 41966029) containing 10% fetal calf serum (FCS, anprotec; AC-SM-0190), 1% penicillin–streptomycin (Gibco 15140–122) at 37 °C with 5% $CO_2$. CH7C17 Jurkat T cells were cultured in Roswell Park Memorial Institute 1640 Medium (RPMI 1640, Gibco, 11875093) supplemented with 10% FCS and 5% HEPES (Roth, 9105.4) at 37 °C with 5% $CO_2$. Expi293F were obtained from Thermo Fisher and cultured in FreeStyle 293 Expression Medium (Thermo Fisher Scientific, 12338001) under agitation at 37 °C with 8% $CO_2$. Cell lines are available from the corresponding authors on request.

### Generation of recombinant bat influenza H18N11 viruses

To generate recombinant cell-culture adapted H18N11 (rP11), the pHW2000-based rescue system was used as described previously[11]. Rescued virus was amplified on MDCK-II cells stably expressing $MHCII_{mEos}$.

### Plasmids

To generate the $MHCII_{mEos}$ plasmid for lentiviral transduction, sequences encoding a signal peptide (MKSLSLLLAVALGLATAVSAGPAV), HLA-DRB1*1501 (NM_001243965.1), mEos3.2 (FPbase ID: VUXRF) and HLA-DRA (NM_019111.4) were amplified and assembled by overlapping fusion PCR. The sequences encoding the influenza model peptide HA307-319 (PKYVKQNTLKLAT) and the PTV-1 2A peptide (ATNFSLLKQAGDVEENPGP) were introduced as overlapping primer overhangs. The assembled construct was cloned into the pLVX-puro vector using BamHI and EcoRI restriction sites. The sequence encoding the 11 amino acid substitution (EIDRYTAIAYW) resulting in $MHCII_{mut}$ was introduced via overlapping fusion PCR. Plasmids encoding $MHCII_{TagRFP}$ and $MHCII_{TagRFPmut}$ for lentiviral transduction were generated analogously. For transient expression of $MHCII_{TagRFP}$ containing the HLA-DRB1*1501 β-chain the sequence was amplified and cloned into the pCAGGS vector using NotI and XhoI restriction sites. The sequences encoding $MHCII_{TagRFP}$ comprising the HLA-DRB1*0101 β-chain (HM067843.1), which is compatible with CH7C17 Jurkat T cells, was synthesized (Azenta Life Science) and cloned into pCAGGS. The correspondent mutant MHCII was generated as described above. Expression plasmid coding for a BlaM1 fusion protein[48] were kindly provided by Silke Stertz. To generate soluble MHCII encoding plasmids, sequences for both the alpha and beta chains were synthesized and cloned into the pCDNA3.1 vector using NotI and EcoRI restriction sites. The alpha chain (HLA-DRA*0101) constructs include a signal peptide (MDWTWRVFCLLAVAPGAHS), a FLAG tag, a Fos zipper domain and a 10xHis-tag, while the beta chain (HLA-DRB1*0101) construct comprises a signal peptide (MVLQTQVFISLLLWISGAYG), a Strep-II tag, the CLIP peptide, an additional FLAG tag and a Jun zipper domain. Sequences were synthesized by Azenta Life Science. Plasmids are available from the corresponding authors on request.

### MHCII surface expression

MDCK-II cells stably expressing $MHCII_{mEos}$ or $MHCII_{mEosmut}$ were seeded in a 12-well plate format. The next day, cells were washed, trypsinized and centrifuged at $15871 \times g$ for 3 min. Cells were then resuspended and incubated with monoclonal anti-HLADR (327002, BioLegend, USA, 1:500) for 30 min on ice. Cells were washed and incubated with APC-labeled anti-mouse (BD Biosciences, 550826, 1:200) for 30 min on ice. Cells were analyzed with a BD LSRFortessa™ flow cytometer (BD Biosciences).

### T cell activation

$8.4 \times 10^5$ cells BHK-21 were seeded in 6-well format and transfected with 1 µg pCAGGS expression vectors encoding $MHCII_{TagRFP}$ or $MHCII_{TagRFPmut}$ comprising the HLA-DRB1*0101 β-chain using Lipofectamine 2000 (Thermo Fisher, 11668027). $MHCII_{TagRFP}$ comprising the HLA-DRB1*1501 β-chain was transfected as negative control. The next day, $5 \times 10^4$ transfected cells were transferred into a well of a 96-well plate and cocultured with $10^5$ CH7C17 Jurkat T cells in RPMI 1640 (Gibco, USA, Gibco, 11875093) supplemented with 10% FCS and 5% HEPES (Roth, 9105.4) for 6 h at 37 °C. Subsequently, cells were stained with FITC-labeled anti-CD3 antibody (BioLegend, 317302, 1:200) and APC-labeled anti-CD69 antibody (Life Technologies, MHCD6905, 1:200) and analyzed with a BD FACSCanto II (BD Biosciences) flow cytometer.

### Susceptibility of MDCK-II cells to infection with H18N11

$MHCII_{mEos}$ or $MHCII_{mEosmut}$ cells were seeded in a 24 well format and infected at an MOI of 5 for 1 h at 37 °C. The inoculum was replaced with growth medium and cells were incubated for 24 h at 37 °C. After incubation, cells were washed, trypsinized and fixed with 4% PFA. Cells were then washed and stained with monoclonal anti-H18 (mouse, produced in-house; 1:100) for 30 min on ice. After washing, cells were stained with and APC-labeled anti-mouse (BD Biosciences, 550826–1:200) for 30 min on ice and analyzed with BD LSRFortessa™ flow cytometer (BD Biosciences)

### M1-β-lactamase entry assay

To generate H18N11 BlaM1 VLPs, HEK293T cells were transfected with pCAGGS-BlaM1 (2 µg), pCAGGS-H18 (0.5 µg) and pCAGGS-N11 (0.5 µg) using Lipofectamine 2000 (Thermo Fisher, 11668027). Supernatant was exchanged 6 h post-transfection with 2 mL of OptiMEM (Gibco, 31985062). Transfected cells were maintained in OptiMEM for 72 h. BlaM1 VLP-containing supernatants were harvested and centrifuged at $1500 \times g$ for 5 min to remove debris. To activate the H18, VLPs were treated with 5 ug/ml TPCK-trypsin (Sigma-Aldrich, 4370285) for 20 min at 37 °C. Inactivation of the trypsin was performed with 10 µg/ml soybean-trypsin-inhibitor (Roche, 10109886001) for 20 min at 37 °C. For detection of cytoplasmic lactamase activity, $10 \times 10^5$ $MHCII_{TagRFP}$ or $MHCII_{TagRFPmut}$ cells were seeded in a 24-well format. Cells were washed and VLPs were added in a total volume of 200 µl and incubated for 4 h at 37 °C. Following incubation, cells were washed, trypsinized and centrifuged at $11363 \times g$ for 3 min. Pellet was then resuspended and loaded with NIR live/dead staining (BioLegend, 423105, 1:1000) and CCF2-AM substrate (ThermoFisher, K1032) for 30 min at 37 °C. After washing, the cell pellet was resuspended in PBS and analyzed with a BD LSRFortessa™ flow cytometer (BD Biosciences).

### Polykaryon formation assay

$8.4 \times 10^5$ HEK293T cells were seeded in 6-well plates and cotransfected with 2 µg of pCAGGS-GFP and either pCAGGS-EV (empty vector) or pCAGGS-H18. For transfection, Lipofectamine 2000 (Thermo Fisher, 11668027) was used at a DNA-to-Lipofectamine ratio of 1:2 in OptiMEM. At 24 h post-transfection, $10^5$ transfected HEK293T cells and $10^5$ MDCK-II cells stably expressing $MHCII_{tagRFP}$ or $MHCII_{tagRFPmut}$, were co-seeded in collagen-coated 24-well plates containing growth medium

(DMEM (Gibco, 41966029)), 10% FCS (FCS, anprotec; AC-SM-0190), 1% penicillin–streptomycin (Gibco, 15140–122) and incubated at 37 °C and 5% $CO_2$. The following day, cells were treated with TPCK trypsin (Sigma-Aldrich, 4370285, 10 µg/ml in OptiMEM) for 30 min at 37 °C. Cells were subsequently washed with PBS, exposed to pH 5 PBS for 20 min at 37 °C and 5% $CO_2$, and then incubated in growth medium for 2 h at 37 °C and 5% $CO_2$. Finally, cells were washed with PBS, fixed with 4% PFA for 20 min, and nuclei were stained in the dark for 1 h using DAPI (Sigma-Aldrich, 28718903, Stock: 1 mg/ml in $H_2O$, 1:10.000) in PBS. Fluorescence images were acquired using a Zeiss Observer.ZI inverted epifluorescence microscope (Carl Zeiss) equipped with an AxioCamMR3 camera using a 20× objective.

## Production of soluble MHCII
sMHCII and sMHCII$_{mut}$ were expressed in Expi293F cells by transfecting 100 µg of plasmid (1:1 ratio of α-chain coding plasmid: β-chain coding plasmid) for 100 ml of cells at $2 \times 10^6$ cells per ml. Transfection was performed using the ExpiFectamine 293 Transfection Kit (Thermo Fisher Scientific, A14524) following the manufacturer instructions. 6 days later, cells were centrifuged, and the supernatant was filtered with a 0.45 µm filter (Sartorius). sMHCII molecules were purified with affinity chromatography using HisPur Cobalt Resin (Thermo Fischer Scientific, 89965) and Strep-TactinXT 4Flow high-capacity resin (IBA life sciences, 2-5030-010), followed by size exclusion chromatography on a Superdex 200 Increase column (Sigma-Aldrich, 28990944) into a buffer containing 20 mM Tris pH 8.0 and 150 mM NaCl.

## Coomassie gel staining
Proteins were separated using SDS-PAGE on a 10% polyacrylamide gel. After electrophoresis, the gel was rinsed briefly with deionized water. The gel was stained with Blauer Jonas (BIOZOL, BZL-GRP1) and incubated for 1 h at room temperature. Following staining, the gel was washed with water to remove the residual solution and was imaged using Vilber E-Box.

## MHCII competition assay
MDCK-II cells stably expressing MHCII$_{mEos}$ were seeded in a 24-well plate format. The next day, sMHCII and sMHCII$_{mut}$ were diluted in MEM medium (Gibco, USA, 31095029) supplemented with 2% FCS, mixed with 50 focus-forming units of H18N11 (rP11) and incubated at 37 °C for 1 h. Cells were pre-treated with PBS containing 15 µg/ml DEAE-Dextran (Sigma-Aldrich, D9885-50G) for 15 min. Subsequently, cells were washed, infected with the H18N11:MHCII mix and incubated for 1 h at 37 °C. Cells were then covered with 500 µL of overlay (DMEM, 20 mM HEPES (Roth, 9105.4) pH 7.5, 200 µg/ml bovine serum albumin (BSA, AppliChem, A1391), 20 µg/ml DEAE-Dextran, 100 µg/ml $NaHCO_3$, 40 µg/ml oxoid agar (Oxoid, LP0028B)) and incubated at 37 °C for 36 h. After overlay removal, cells were fixed with 4% PFA for 20 min, permeabilized with PBS, 0.1% Triton and blocked with PBS containing 5% FCS. Cells were then stained with a monoclonal anti-H18N11-NP antibody (produced in-house, 1:500) for 1 h at room temperature, washed and incubated with a secondary HRP-conjugated anti-mouse antibody (Dianova, 115546062, 1:500) for 1 h at room temperature. After washing, focus forming units were visualized by adding the substrate solution (PBS supplemented with 0.5 µg/ml 3, 3′-diaminobenzidine (DAB, Sigma-Aldrich, D12384-1G), 0.5 µg/ml nickel ammonium sulfate (Alfa Aesar, A18441.22) and 0.015% $H_2O_2$ (Sigma-Aldrich, 7722-84-1).

## Quantification of MHCII nanoclustering via PALM in fixed cells
To determine the nanoscale organization of MHCII within the plasma membrane, MDCK-II cells stably expressing MHCII$_{mEos}$ or MHCII$_{mEosmut}$ were grown on 25 mm glass cover slips overnight, then fixed in PFA 4% for 15 min at room temperature. Glass cover slips were mounted in an AttoFluor cell chamber (ThermoFisher), for imaging. PALM was performed on a Nikon Ti Eclipse NSTORM microscope

equipped with four laser lines at 405 nm (Coherent), 488 nm (Sapphire, Coherent), 561 nm (Sapphire, Coherent) and 642 nm laser (F-04306-113, MPB Communications). The sample was observed through a Nikon Apo TIRF 100x Oil DIC N2 NA 1.49 objective and emitted light was detected on an Andor iXon3 DU-897 EMCCD camera (Oxford Instruments). The width of a square camera pixel corresponds to 160 nm on the sample. mEos3.2 was photo-converted with a 405 nm laser (power at the objective back aperture of 0.1–5 µW) and excited with 561 nm laser (power at the objective back aperture of 0.1–2 mW) or with a 488 nm laser (power at the objective back aperture of 0.1–2 mW). Emitted light was directed onto the camera using a dichroic mirror and filtered through a Bright Line HC 609/64 emission filter. The emission of mEos3.2 was thus detected in between 577 and 641 nm. We typically took 20k frames at 30 ms integration time using the NIS elements software (AR 5.21.03 64-bit). Single emitter positions were localized using the software DECODE, the processed using custom MatLab (Version R2020b, MathWorks) routines as described previously[21,49]. Lateral sample drift was corrected via redundant cross-correlation using Thunderstorm v1.3 (Ref). Localizations were clustered using the build-in DBSCAN function of MatLab R2020b (Mathworks). Images were rendered and cropped for visualization using Thunderstorm and Fiji. Plotting and statistical analysis were performed using Prism 9 (GraphPad). Image resolution was determined using Fourier Ring Correlation (FRC) and image decorrelation analysis[50,51]. To then investigate the MHCII clustering in the presence of the virus, MHCII$_{mEos}$ or MHCII$_{mEosmut}$ were incubated on ice for 1 h with H18N11, at an MOI of 200. Unbound viruses were washed away with cold PBS and the cells were fixed with PFA 4% for 15 min at room temperature. For the staining, the cells were incubated for 1 h in blocking buffer (0.2% BSA in PBS). Primary polyclonal anti-H18 (rabbit, produced in-house) was diluted 1:750 in blocking buffer and the cells were incubated for 1 h at room temperature. The cells were then washed 3 times in PBS and incubated for 1 h at room temperature with secondary antibody anti-rabbit AlexaFluor647 (ThermoFisher, 1:250). Finally, the cells were washed 3 times in PBS. Images were acquired as described above (20k frames per acquisition).

## Single-virus tracking
$4.5 \times 10^5$ MHCII$_{TagRFP}$ or MHCII$_{TagRFPmut}$ were seeded on a 35 mm² dish (Ibidi). The next day, cells were pre-incubated for 15 min on ice. Growth medium was discarded and replaced with infection medium containing DiO (ThermoFisher, V22886)-labeled H18N11 filtered using a 0.2 µm pore size sterile filter. Cells were incubated with virus on ice for 30 min. Cells were then washed with cold PBS and covered with pre-warmed growth medium. The samples were imaged at 37° and 5% $CO_2$ using a Nikon CSU-W1 spinning disk confocal microscope for 20 min. The recorded movies were analyzed using TrackMate and the trajectories further processed using MatLab R2020b (Mathworks).

## Measurement of MHCII single protein diffusion using sptPALM in live cells
H18N11 virus was diluted 1:1 in PBS and labeled with DiD (Thermo-Fisher, V22887) for 30 min at room temperature in the dark. Viral particles were separated from unbound dye by gel filtration (Nap5 columns, equilibrated with PBS) (Cytiva, 17085201). The labeled virus particles were then vortexed, spun down and filtered over a 0.22 µm pore size filter and subsequently resuspended in PBS. 3D-Epoxy Glass Coverslips (PolyAN, 10400206) were mounted at the bottom of a 6-well chamber (ibidi, 80828). These slides allow covalent immobilization by reacting with nucleophilic groups. The DiD labeled virus was added and mixed with primary buffer (150 mM $Na_2HPO_4$, 50 mM NaCl, pH 8.5) and incubated overnight at 4 °C. The following morning, the unbound viruses were washed away and the secondary buffer (50 mM ethanolamine, 100 mM Tris, pH 9.0) was added for 1 h at room temperature. Wells were then washed first with PBS and then with infection

medium (DMEM, 0.2% BSA, penicillin-streptomycin). $3 \times 10^4$ MHCII$_{mEos}$ or MHCII$_{mEosmut}$ cells were seeded in each well and cultured until fully attached, typically overnight. MHCII molecules were imaged via PALM using the optical microscope setup described above but at 37 °C and 5% CO$_2$. The sample was excited with a 642 nm laser (DiD) with a laser power at the objective back aperture of 5–15 mW. The PALM image stacks were taken using an Andor iXon 897 EMCCD camera with an EM gain of 200 and an integration time of 30 ms/frame. Under live cell conditions, we typically acquired 10k frames and detected between 10-100 localizations per frame using the NIS elements software (AR 5.21.03 64-bit).

Single emitter positions were localized using the software DECODE[49] and trajectories were reconstructed using the TrackPy Python library. Localizations were linked using a KDTree algorithm with a maximum search range of 0.8 pixels and a zero-frame-gap memory. The adaptive stop for solving oversized subnets was set to 0.1 with a step size multiplicator of 0.9. For further analysis, only tracks with a minimum length of 10 frames were considered. Ensemble drift xy(t) was calculated and subtracted from the tracks. Mean squared displacement (MSD, equation 1) was determined and plotted individually for all molecules. For all tracks, the initial diffusion coefficients (D$_{1-3}$) and the power law exponent alpha were calculated by applying a linear fit in log space (Equation 2). Power law exponents and diffusion coefficients for virus spots were calculated within an $800 \times 800$ nm square around an immobilized virus.

$$MSD(T) = \frac{1}{T}\sum_{t=1}^{T}(x(t) - x_0)$$

$$MSD(T) = 4Dt^\alpha$$

### Staining of single viral particles
For NP and H18 staining, 50 µl of H18N11 virus were diluted in 250 µl primary buffer (150 mM Na$_2$HPO$_4$, 50 mM NaCl, pH 8.5) and incubated up to 4 h at 4 °C on 3D-Epoxy Slides (PolyAN, 10400201). Unbound virus was washed with PBS. Following washing, bound viral particles were fixed for 10 min at room temperature in 4% PFA and washed twice. For staining, fixed particles were incubated for 1 h at room temperature in permeabilization-blocking buffer (PBS containing 5% BSA and 0.2% TritonX-100). Primary polyclonal anti-H18 (rabbit, produced in-house, 1:750) and primary monoclonal anti-H18N11-NP antibody (mouse, produced in-house, 1:500) were diluted in permeabilization-blocking buffer and incubated overnight at 4 degrees. The following morning the particles were washed 3 times in PBS and incubated for 1 h at room temperature with secondary antibody anti-rabbit AlexaFluor647 (ThermoFisher, 1:1000) and Alexa-Fluor488 (Dianova, 115546062, 1:1000). Finally, the particles were washed 3 times in PBS and once in MilliQ water and mounted with antiface mountant (Invitrogen, P36965). For DiD and H18 staining, the virus was labeled and immobilized on 3D-Epoxy Glass Coverslips, as previously described. Following the incubation in primary buffer, the bound viral particles were washed and fixed in 4% PFA at room temperature for 10 min. For H18 staining, fixed particles were incubated for 1 h at room temperature in blocking buffer (PBS containing 5% BSA). Primary polyclonal anti-H18 (rabbit, produced in-house, 1:750) was diluted in blocking buffer and incubated overnight at 4 degrees. The following morning the particles were washed 3 times in PBS and incubated for 1 h at room temperature with secondary antibody anti-rabbit AlexaFluor488 (Invitrogen, A21441, 1:1500). Finally, the particles were washed 3 times in PBS. For all the samples, structured illumination microscopy (SIM) imaging was performed with an ELYRA 7 microscope (Zeiss) equipped with a Plan-Apochromat 63x/1.4 Oil DIC

M27 objective. Acquired imaging data were SIM processed using Zen 3.0 black edition and analyzed with Imaris 10.2.0

### Transmission electron microscopy of viral particles
To visualize virus particles via the negative stain method, a thin carbon support film was floated on a droplet of virus solution to allow adsorption of particles for ~45 s. Afterwards, the carbon was bound to a copper grid (mesh 300), washed twice on water droplets and finally put on a drop of 4% (w/v) aqueous uranyl acetate, pH 5.0, for 45 s. The exogenous liquid was taken away with a filter paper and the grid heat-dried on a light bulb. Samples were examined in a Zeiss Libra120 Plus transmission electron microscope (Carl Zeiss, Oberkochen, Germany) at an acceleration voltage of 120 kV at calibrated magnifications using ITEM Software (Olympus Soft Imaging Solutions, Münster).

### Reporting summary
Further information on research design is available in the Nature Portfolio Reporting Summary linked to this article.

## Data availability
Source data are provided with this paper. Example datasets are available via https://zenodo.org/records/14251817 or via https://doi.org/10.5281/zenodo.14251817. Source data are provided with this paper.

## Code availability
Newly generated, custom-written MatLab and Python scripts are available at https://github.com/christian-7/NIBI/tree/main/code_Broich_et_al or via https://doi.org/10.5281/zenodo.15005579.

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

## Acknowledgements

We thank Wolfgang Schamel for providing the CH7C17 Jurkat T cells and Mathias Müsken (HZI) for help with electron microscopy. We are grateful to Silke Stertz for the m1-Bla expression plasmid. This work was supported by grants from the European Research Council (ERC) to M.S. (NUMBER 882631—Bat Flu) and in part by the Excellence Initiative of the German Research Foundation (GSC-4, Spemann Graduate School) and the Ministry for Science, Research and Arts of the State of Baden-Wuerttemberg. M.K.O and J.R. are members of the Spemann Graduate School of Biology and Medicine (SGBM). A.G.W. was supported by the Wellcome Trust grant 303026/Z/23/Z and by core funding from the Wellcome Trust (CC2060), Medical Research Council UK (CC2060), and Cancer Research UK (CC2060) at the Francis Crick Institute. P.R. was supported by the Hans A. Krebs Medical Scientist Programme of the Medical Faculty of the University of Freiburg. C.S. acknowledges support by the Helmholtz Association (VH-NG-1526). The funders had no

role in study design, data collection and analysis, decision to publish, or preparation of the manuscript. The work was supported by a major research instrumentation grant to R.G. (INST 39/1170-1 FUGG). The Lighthouse Core Facility is funded in part by the Medical Faculty, University of Freiburg (Project Numbers 2023/A2-Fol; 2021/A2-Fol; 2021/B3-Fol) and the DFG (Project Number 450392965).

## Author contributions

Conceptualization: M.K.O, J.R., R.G., M.S., C.S., and P.R.; Conduction of experiments: M.K.O, J.R., and L.B.; Data analysis: M.K.O, J.R., L.B., D.F., R.G., A.G.W., K.C., C.S., and P.R.; Writing and original draft preparation: C.S. and P.R.; Review and editing: M.K.O, J.R., L.B., R.G., A.G.W., K.C., and M.S.; Supervision and project administration: C.S. and P.R.

## Funding

## Competing interests

The authors declare no competing interests.
