## [Transparent Peer Review file · Nature Communications]

The bat Influenza A virus subtype H18N11 induces nanoscale MHCII clustering upon host cell attachment.

Corresponding Author: Dr Peter Reuther

Version 0:

Reviewer comments:

Reviewer #1

(Remarks to the Author)

In this work Kaukab Osman et al, propose a new receptor for H17 and H18 (MHC II); furthermore they hypothesize that bat HA would likewise interact with multiple MHCII molecules and also that these molecules constitute the main receptors for this IAV variants. Conventional IAV bind sialic acid microclusters with low affinity, followed by endocytosis and fusion in late endosomal compartments. Similarly and with super-resolution PALM microscopy, the authors propose that IAV from bats also induce microclusters of MHC II (they also see that the viruses change motility when engaging with pre-existing microclusters).

Overall, the findings reported in this study, and the advanced technologies in super-resolution imaging are of interest.

The validation of labelled MHC II complex in T cells together with the IAV fusion assays (Figure 1) is convincing. However these experiments alone do not necessarily mean MHC II receptors are key for IAV fusion alone. For the BlaM assays, could the authors perform as a negative control a IAV virus (which contains M1-BIAM) without H18? This would set the threshold for no fusion at all and will add a reference to understand if the big diminution seen with the MHC mutant is significant at all or not.

The neutralization experiments (Figure 2) employing soluble MHC II look very convincing to me. Showing the importance of this ligand in entry and fusion.

For the PALM experiments (Figure 3) in fixed cells, it would have been nice to employ labelled IAV particles (M1-mCherry?) so that we can relate the H18 data (employing antibodies) and the real viral particles. These signals could come from exosomes and large aggregates, and therefore the establishment of high microclusters of MHC II around the virus needs to be corroborated identifying productive particles from potential junk (which is present in high amounts in virus production protocols).

The same criticism applies to (Figure 4) DiD is a lipophilic dye that will label every particle (comprised junk and exosomes) that might contain or not H18 or other IAV receptors. It is crucial that the authors identify an efficient way to label their IAV particles if they want to quantitatively establish this type of assays. I suggest labelling both the trimer and a matrix capsid. This way, one can probe IAV H18 – MHC II interactions with real particles. It is unclear, the way the figures are shown and experiments prepared what it is truly the authors are imaging.

Reviewer #2

(Remarks to the Author)

The papers Christian Sieben describe the nanoscale clusters of influenza A virus receptors upon HA binding.

The data is compelling and of interest to a broad readership. The microscopy approaches are fantastic. However, the first manuscript is of a technical complexity, probably beyond a part of that broad readership. The capture of live cells using both virus and fibronectin comes across more as a trick, without real biological consequences. Would there be fluorescently

labelled glycosylated membrane proteins that are not clustered upon engagement of the virus?

This methodology was then applied to H18N11 in an accompanying paper. This virus uses a protein as a receptor and thus rules of engagement could be different, but in the end was not. In this paper the interaction of H18 and different MHC construct where first studied in detail before state-of-the-art microscopy approaches were used to dissect receptor clustering. Due to the availability of a null-mutant receptor protein clustering was convincing.

Eventually clustering of receptors appears to be quite universal, and this could be discussed more. It also decreases the novelty somewhat as it is quite expected and the technological advancements is then the novelty here.

Single IAVs induces Nanoscale cellular reprogramming at the virus-cell interface

Major

-The introduction steers towards a protein receptor, yet these are proteins that migrate from and to the plasma membrane, sialylated glycans on those proteins are the receptors. This change in premise should be accounted for.

-Fig 1D, sure viruses are located under the cell, but also next to it, there is no data for actual attachment as this could just be due to the fibronectin peptide (which by itself is a smart move). The concluding text of this paragraph should be toned down.

-The same holds true for the conclusion of the next "chapter", it is not known if this cluster is "functional". Again, probably glycans on EGFR are bound so it should say capable of clustering EGFR or something along that line.

-Only 15% of the viruses colocalize with actin? Is this a lot? What does this mean? It is gleaned over it, I also don't get the arc, blobs and holes classification.

Minor

-IAV viral envelope proteins are not spikes...

-sialylated glycans are receptors and not attachment factors, please change throughout

-the graphs Fig 1 B and C should be discussed and referenced in the text better, I had to go back and forth many times.

-Why would you compare fluorescent beads with fluorescent beads on glass surfaces?! Just that the coating is efficient?

-Because AP2 is recruited it can also still be dispensable, so this is not so surprising.

-First line of the discussion recruitment of glycoproteins

The bat IAV H18N11 induces nanoscale MHCII clustering upon host cell attachment

Major

- "inverse infection" while there is no infection, just binding...

-Was the addition of the fibronectin peptide necessary here as well?! This should be described in an accompanying paper.

Minor

-discussion, bat and avian IAV both cluster a receptor for endocytosis that is conserved for other viruses as well... this could be further discussed

-hardly any information is given in the methods for the creation of plasmids for the soluble expression of MHC proteins

-same goes for the inverse binding assay how were these viruses coated the glass slides and how was this controlled?

Reviewer #3

(Remarks to the Author)

Review Report:

The paper titled, "The bat Influenza A virus subtype H18N11 induces" by Osman et al., studies the MHCII clustering and the attachment of IAV particles to MHCII clusters. The authors used PALM based super-resolution studies on fixed and live cells to validate the study. The study is performed on immobilized viral particles on coverslips followed by cell seeding on the coverslip. In this paper, super-resolution studies is the only novel feature. Here are my major comments:

1. The main weakness of the manuscript is the lack of time-lapse super-resolution imaging. This study has the ability to show the cluster formation in live cell and would be a direct proof for cluster formation.
2. There is no transmission and widefield fluorescence image (post expression) of the cell in Fig. 3 and 4. Normally, this is done by imaging cells in transmission mode (available with most of the modern inverted microscopes) and adding a separate fluorescence optical arm to the existing PALM system.
3. Small details of the PALM microscopy such as optical components and the emission range / spectra of the photo-convertible fluorophore is missing in the main manuscript.
4. Line 186-188: The statement, "This analysis reveals This suggests an MHCII ... cell surface" needs further justifications based on single molecule analysis.
5. Studies carried out in the literature related to super-resolution study of Influenza A virus is completely ignored. Please see the following papers that clearly indicate clustering of HA in cellular system.
 - Nikki M. Curthoys et al., Influenza Hemagglutinin Modulates Phosphatidylinositol 4,5-Bisphosphate Membrane Clustering, *Biophys J* . 116, 893-909 (2019).
 - S T Hess et al., Dynamic clustered distribution of hemagglutinin resolved at 40 nm in living cell membranes discriminates

between raft theories, PNAS 104, 17370-17375 (2007).

- Raut P, Obeng B, Waters H, Zimmerberg J, Gosse JA, Hess ST., Phosphatidylinositol 4,5-Bisphosphate Mediates the Co-Distribution of Influenza A Hemagglutinin and Matrix Protein M1 at the Plasma Membrane. *Viruses*. 14, 2509 (2022).

6. A lot of information is missing in the reported super-resolution study such as, localization plot, Fourier Ring Correlation, drift-correction etc.

7. Super-resolution data is noisy, and does not show stable clusters.

8. The clustering is performed on fixed cells. I miss time-lapse super-resolution imaging data showing dynamic cluster formation. It is important to show super-resolved images at different time points in a live cell to demonstrate formation of clusters.

9. The characteristics of MHCII cluster such as, cluster density, number of MHCII per cluster and fraction of MHCII participating in cluster formation process need to be determined. This has a direct link to the reported study related to cluster formation and physiological state of the cells.

10. Finally, I feel more work needs to be done for a complete study and the work better suits core virology journals.

Overall, the work lacks cluster analysis, determination of critical parameters related MHCII clustering, and ignores reported super-resolution study on Influenza A virus. In general, more work needs to be done for a complete conclusive study.

Version 1:

Reviewer comments:

Reviewer #1

(Remarks to the Author)

The authors have provided a negative control for the Blame assay.

Even if the IAV particles were produced with ultracentrifugation, these samples might be prone to contain exosomes that are not IAV particles but engage with the cell. In the accompanying paper the authors produced a figure (EM) that shows that some of these particles were in fact not viral particles at all.

Overall, this is a good paper and deserves to be published.

Reviewer #2

(Remarks to the Author)

The authors sufficiently replied to my suggestions / comments

Reviewer #3

(Remarks to the Author)

My Comments:

On the bright side, the cluster parameters are now provided. However, still a lot of other information is missing. See my comments below:

1. Fig. 4d-f is not time-lapse super-resolved reconstruction at varying time-points. Fig. 4d-f does not show the formation of clusters as claimed in the study. Moreover, Supplementary figure 4k-l is not the time-lapse image of the entire cell. Ideally, time-lapse imaging of an entire cell is likely to give visual evidence of cluster formation with time. This will be the conclusive proof.

2. Supplementary figure 3 and 4 does not contain light transmission image. Moreover, the images quality is not good. Specifically, the representative widefield images of H18N11 viral particles (magenta) and MDCK-II cells are of poor quality. Drawing any conclusion from the imaging data is still doubtful.

3. I still miss details related to the imaging system. For example, what is the power used for photoactivation and excitation of the mentioned lasers (405, 488, 561, 642nm). Moreover, information related to dwell-time, EMCCD gain, average number of molecules detected per frame and other details are missing.

4. Cluster identification in Supplementary Fig. 5 does not seem to be accurate. One can see lot more clusters than what is identified by the algorithm. Re-check chosen parameters in the cluster algorithm.

5. Still, the super-resolved images are very noisy.

Version 2:

Reviewer comments:

Reviewer #1

(Remarks to the Author)

I was happy with the previous version of the manuscript and I have now read the concerns from another reviewer and also the answers from the authors.

In terms of the quality of the images, the analysis of clusters in cells and also the use of bright-field I consider that all these questions were correctly answered by the authors.

The new figures, and clarifications are satisfactory to me.

We would like to thank all reviewers for their constructive feedback and provide a point-by-point response to the individual comments below.

Original Reviewers' comments are in black.

Our response is in blue, changes in the main manuscript are highlighted in yellow.

Reviewer #1 (Remarks to the Author):

In this work Kaukab Osman et al, propose a new receptor for H17 and H18 (MHC II); furthermore they hypothesize that bat HA would likewise interact with multiple MHCII molecules and also that these molecules constitute the main receptors for this IAV variants. Conventional IAV bind sialic acid microclusters with low affinity, followed by endocytosis and fusion in late endosomal compartments. Similarly and with super-resolution PALM microscopy, the authors propose that IAV from bats also induce microclusters of MHC II (they also see that the viruses change motility when engaging with pre-existing microclusters).

Overall, the findings reported in this study, and the advanced technologies in super-resolution imaging are of interest.

The validation of labelled MHC II complex in T cells together with the IAV fusion assays (Figure 1) is convincing. However these experiments alone do not necessarily mean MHC II receptors are key for IAV fusion alone. For the BlaM assays, could the authors perform as a negative control a IAV virus (which contains M1-BIAM) without H18? This would set the threshold for no fusion at all and will add a reference to understand if the big diminution seen with the MHC mutant is significant at all or not.

Response: The control suggested by this reviewer would be valuable, however, generation of IAV VLPs without HA, unfortunately is not possible. Nevertheless, in order to set the threshold for no fusion at all, we now included wildtype MDCK-II cells. These cells do not express MHCII and are therefore resistant to infection with bat IAV (Comp. Karakaus et al. 2019, <https://doi.org/10.1038/s41586-019-0955-3>). Despite the fact that MHCII expression is a prerequisite for infection with bat IAV, we agree with the reviewer that there might be additional, critical entry factors, which yet remain to be discovered. As shown in the revised panel Fig 1f, b-lactamase activity in wildtype MDCK-II cells is indistinguishable from cells expressing the MHCII mutant. This is further highlighted in the new supplementary panel S1e where we now show exemplary FACS plots.

The neutralization experiments (Figure 2) employing soluble MHC II look very convincing to me. Showing the importance of this ligand in entry and fusion.

For the PALM experiments (Figure 3) in fixed cells, it would have been nice to employ labelled IAV particles (M1-mCherry?) so that we can relate the H18 data (employing antibodies) and the real viral particles. These signals could come from exosomes and large aggregates, and therefore the establishment of high microclusters of MHC II around the virus needs to be corroborated identifying productive particles from potential junk (which is present in high amounts in virus production protocols).

The same criticism applies to (Figure 4) DiD is a lipophilic dye that will label every particle (comprised junk and exosomes) that might contain or not H18 or other IAV receptors. It is crucial that the authors identify an efficient way to label their IAV particles if they want to quantitatively establish this type of assays. I suggest labelling both the trimer and a matrix capsid. This way, one can probe IAV H18 – MHC II interactions with real particles. It is unclear, the way the figures are shown and experiments prepared what it is truly the authors are imaging.

Response: The reviewer raises an important concern. In our laboratory, we have extensively explored different possibilities to introduce reporter genes into the IAV genome (e.g. <https://doi.org/10.1038/srep11346>). Despite several attempts, however, we were not able to generate an infectious IAV encoding a fluorescently labeled M1.

Due to the fact that the virus preparations we used for this study were purified by ultracentrifugation, we assumed that they were pure and largely free of cellular debris. Yet, in order to test for the possibility that there are H18-containing exosomes or large aggregates in our virus preparation, which might have confounded our analysis, we performed a thorough quality control (Supplementary movies 5 and 6, Fig. S6). We immobilized our virus stock on epoxy-slides and stained for H18 and NP. In an analogous approach, we combined DiD-membrane labeling with the H18 staining. As shown in the new supplementary panel S6a and S6b, the majority of viral particles in our preparation stain positive for both H18 and NP or DiD and H18. In addition, we subjected our virus stock to electron microscopy (S6c). This analysis revealed that our virus preparation consists of pleomorphic particles that resemble bona fide bat IAV virions (Comp. Moreira et al. 2016: <https://doi.org/10.1073/pnas.1608821113>). Taken together, we hope that this quality controls convince this reviewer and the readership that our virus preparation is rather homogeneous and that the H18 : MHCII interaction observed in our study really derives from an interaction with authentic viral particles.

Reviewer #2 (Remarks to the Author):

The papers Christian Sieben describe the nanoscale clusters of influenza A virus receptors upon HA binding.

The data is compelling and of interest to a broad readership. The microscopy approaches are fantastic. Although the first manuscript is of a technical complexity, probably beyond a part of that broad readership. The capture of live cells using both virus and fibronectin comes across more as a trick, without real biological consequences. Would there be fluorescently labelled glycosylated membrane proteins that are not clustered upon engagement of the virus?

Response: We thank the reviewer for this assessment. We would like to clarify the use of fibronectin in both manuscripts since there might have been some confusion. Broich *et al.* describe a glass slide preparation in which viruses and cRGD are immobilized during the same step via reactive NHS. cRGD, a small circular peptide derived from fibronectin, was important to allow efficient attachment of cells to our otherwise pegylated glass surface (Broich *et al.*: Fig. 1d). In the paper by Osman *et al.*, we also started to use commercial slides functionalized with a 3D epoxy surface (PolyAn, Berlin). Epoxy residues can form covalent bonds with biomolecules and we found that here cRGD was not required for efficient cell attachment. Presumably, this is due to the underlying polymer matrix used to generate the 3D epoxy surface. Importantly, after virus immobilization, all remaining reactive epoxy groups are quenched before cells are added. Hence, cellular proteins are not immobilized by this approach. We have now added additional information in the methods section to clarify this point (Broich *et al.*, lines 459-462). The reviewer is correct that we did not observe any differences between the two virus immobilization approaches.

We expect that fluorescently-labelled glycosylated membrane proteins that are e.g. differently localized at the cell membrane than EGFR are not clustered upon engagement of the virus. This is currently analyzed in a larger ongoing study and beyond the scope of this study.

This methodology was then applied to H18N11 in an accompanying paper. This virus uses a protein as a receptor and thus rules of engagement could be different, but in the end was not. In this paper the interaction of H18 and different MHC construct where first studied in detail before state-of-the-art microscopy approaches were used to dissect receptor clustering. Due to the availability of a null-mutant receptor protein clustering was convincing.

Eventually clustering of receptors appears to be quite universal, and this could be discussed more. It also decreases the novelty somewhat as it is quite expected and the technological advancements is then the novelty here.

Response: We agree with this conclusion and have extended the discussion (Broich *et al.*, lines 356-357).

Single IAVs induces Nanoscale cellular reprogramming at the virus-cell interface

Major

-The introduction steers towards a protein receptor, yet these are proteins that migrate from and to the plasma membrane, sialylated glycans on those proteins are the receptors. This change in premise should be accounted for.

Response: We have modified the manuscript accordingly, now using the term *receptor* for the sialylated glycans.

-Fig 1D, sure viruses are located under the cell, but also next to it, there is no data for actual attachment as this could just be due the the fibronectin peptide (which by it self is a smart move). The concluding text of this paragraph should be toned down.

Response: We agree that this could be a possibility and have thus modified the 2nd paragraph on page 5 (lines 155-159).

-The same holds true for the conclusion of the next “chapter”, it is not known if this cluster is “functional”. Again, probably glycans on EGFR are bound so it should say capable of clustering EGFR or something along that line.

Response: We agree and have modified the paragraph 3 on page 7 (lines 229-231).

-Only 15% of the viruses colocalize with actin? Is this a lot? What does this mean? It is gleaned over it, I also don't get the arc, blobs and holes classification.

Response: We apologize for not providing more details about our conclusions. From the STORM data of phalloidin-stained fixed cells, we classified the organization of filamentous actin at the virus-binding site into three groups 1) local enrichment (i.e., blob), 2) absence of signal (i.e., hole) or 3) ring- or arc-like structures. The live-cell imaging then revealed similar structures resembling these three groups, suggesting that they might be part of the same actin filament turnover process. Since the actin re-organization was very dynamic, as shown in Figure 4, the local enrichment was short-lived and only visible in 10-20% of the frames in the live cell movie. This observation is thus consistent with our observation in fixed cells considering a larger number of viruses and assuming similar actin turnover.

We thank the reviewer for this comment and have now extended the respective discussion in paragraph 1 on page 10 (lines 305-309).

Minor

-IAV viral envelope proteins are not spikes...

Response: We modified the wording accordingly.

-sialylated glycans are receptors and not attachment factors, please change throughout

Response: We modified the wording accordingly as mentioned above.

-the graphs Fig 1 B and C should be discussed and referenced in the text better, I had to go back and forth many times.

Response: We added references in paragraph 2 on page 4, paragraph 1 on page 5 (lines 130-131 and 136-137).

-Why would you compare fluorescent beads with fluorescent beads on glass surfaces?! Just that the coating is efficient?

Response: There might have been some confusion here, but we compared fluorescent beads with DID-labelled IAVs. This was to test the efficiency of the coating and the degree of particle aggregation in our samples and during immobilization. We looked at the intensity distribution and found that both samples show a single overlapping peak indicating that virus particles did not aggregate during the labelling and immobilization procedure.

-Because AP2 is recruited it can also still be dispensable, so this is not so surprising.

Response: We agree, from the presence of AP2 at the viral attachment site, we cannot conclude that IAVs rely on this adaptor for CME as other adaptors were also shown to be involved. However, the recruitment of AP2 with dynamics resembling that of AP2-dependent CME indicates a positive involvement. But this needs to be tested for instance in AP2 knock-out cells. We have thus modified the wording accordingly (lines 346-348).

-First line of the discussion ◊ recruitment of glycoproteins

Response: We modified the wording accordingly.

The bat IAV H18N11 induces nanoscale MHCII clustering upon host cell attachment
Major

- "inverse infection" while there is no infection, just binding...

Response: We now changed the name of the method to "inverted attachment".

-Was the addition of the fibronectin peptide necessary here as well?! This should be described in an accompanying paper.

Response: We apologize for the confusion. As described in the response to comment 1, we have used two different virus immobilization protocols in the two studies.

Minor

-discussion, bat and avian IAV both cluster a receptor for endocytosis is conserved for other viruses as well... this could be further discussed

Response: We now discuss this and propose adaptation of the "inverse attachment" approach in order to study other virus-receptor interactions (lines 271-276).

-hardly any information is given in the methods for the creation of plasmids for the soluble expression of MHC proteins

Response: We now describe the generation of the MHC-II expression plasmids in more detail in the material and method section.

-same goes for the inverse binding assay how where these viruses coated the glas slides and how was this controlled?

Response: We added a more detailed description of the coating to the respective chapter in the “material and methods” section.

Reviewer #3 (Remarks to the Author):

Review Report:

The paper titled, “ The bat Influenza A virus subtype H18N11 indices ” by Osman et al., studies the MHCII clustering and the attachment of IAV particles to MHCII clusters. The authors used PALM based super-resolution studies on fixed and live cells to validate the study. The study is performed on immobilized viral particles on coverslips followed by cell seeding on the coverslip. In this paper, super-resolution studies is the only novel feature. Here are my major comments:

1. The main weakness of the manuscript is the lack of time-lapse super-resolution imaging. This study has the ability to show the cluster formation in live cell and would be a direct proof for cluster formation.

Response: We agree with the reviewer on the importance of live-cell imaging, but want to point out that we have performed extensive live-cell super-resolution imaging as presented in Figure 4 d-f. The figure shows live-cell single-particle tracking (spt)PALM data of intact viruses interacting with live cells and represents one novel application of our virus immobilization approach. We see this as a particular strength to visualize virus-receptor interaction at the single-molecule level.

To better visualize the dynamics of MHCII at the virus-binding site, we have now rendered our sptPALM data in time-binned time-lapse super-resolution movies. The movies are now included as Supplementary Movies 1-4 showing the dynamics of MHCII and MHCII^{mut} at the H18 virus binding site. The individual frames and their intensity quantification are now also shown in the new Figure S4k-n. The dynamic response of MHCII clusters can be observed, indicating the exchange of MHCII between the virus interface and the remaining plasma membrane. We indeed observe and now show in Figure S4k-l that some viruses were already associated with MHCII clusters at the beginning of the acquisition but also the appearance/formation of new clusters (marked with yellow triangles) later during the acquisition period. From this data, we conclude that clusters indeed form at the virus-binding site.

2. There is no transmission and widefield fluorescence image (post expression) of the cell in Fig. 3 and 4. Normally, this is done by imaging cells in transmission mode (available with most of the modern inverted microscopes) and adding a separate fluorescence optical arm to the existing PALM system.

Response: As the reviewer suspected, we also always acquire WF images to document the respective field of view. We have now included WF images of the respective imaging scenarios in the new Figures S3a-b and S4i-j.

3. Small details of the PALM microscopy such as optical components and the emission range / spectra of the photo-convertible fluorophore is missing in the main manuscript.

Response: We have added more details of the optical setup in the methods sections and provide information and the respective fluorescent protein database entry of the used photo-convertible fluorescent protein.

4. Line 186-188: The statement, “This analysis reveals This suggests an MHCII ... cell surface” needs further justifications based on single molecule analysis.

Response: Following the reviewer’s suggestion (Point 9), we now also analyzed the MHCII cluster density, resulting in a lower density for the mutant (new Fig. S3d). We modified our statement accordingly (lines 198-201).

5. Studies carried out in the literature related to super-resolution study of Influenza A virus is completely ignored. Please see the following papers that clearly indicate clustering of HA in cellular system.

- Nikki M. Curthoys et al., Influenza Hemagglutinin Modulates Phosphatidylinositol 4,5-Bisphosphate Membrane Clustering, *Biophys J* . 116, 893-909 (2019).
- S T Hess et al., Dynamic clustered distribution of hemagglutinin resolved at 40 nm in living cell membranes discriminates between raft theories, *PNAS* 104, 17370-17375 (2007).
- Raut P, Obeng B, Waters H, Zimmerberg J, Gosse JA, Hess ST., Phosphatidylinositol 4,5-Bisphosphate Mediates the Co-Distribution of Influenza A Hemagglutinin and Matrix Protein M1 at the Plasma Membrane. *Viruses*. 14, 2509 (2022).

Response: We now cite amongst others the 3 studies suggested by this reviewer (line 92-93).

6. A lot of information is missing in the reported super-resolution study such as, localization plot, Fourier Ring Correlation, drift-correction etc.

Response: We thank the reviewer for raising this point. Some information here was indeed missing. We now present localization and clustering plots in the new Figures S5 and S7 together with the respective resolution estimates. Here, we use Fourier Ring Correlation (FRC) as suggested by the reviewer but also image decorrelation (Descloux, *Nat Meth*, 2019), a more recent parameter-free method for estimating the resolution in diverse imaging datasets. New Figure S5 shows an example of MHCII-mEos3.2 and MHCIImut-mEos3.2 representative for the fixed-cell PALM data shown in Figure 3. New Figure S7 shows the resolution estimation of the live-cell sptPALM imaging of MHCII-mEos3.2 and MHCIImut-mEos3.2 imaged using TIRF. Lateral sample drift was corrected routinely as part of our standard localization processing routine using redundant cross-correlation (RCC) developed by Wang, *Optics Express*, 2014. We have now added more details about the drift correction procedure in the material and methods section.

7. Super-resolution data is noisy, and does not show stable clusters.

Response: We agree and have thus used a clustering threshold of 30 localizations to avoid noisy or spurious clusters. The clustering statistics, now presented in Figure S3c-e, revealed that MHCII clusters contain on average 119 localizations, MHCIImut cluster 37 localizations. In addition, the time-lapse dynamics of MHCII and MHCIImut, now presented in Figure S4k-l show clusters that are well visible above the noise, which is largely averaged out during the time binning.

8. The clustering is performed on fixed cells. I miss time-lapse super-resolution imaging data showing dynamic cluster formation. It is important to show super-resolved images at different time points in a live cell to demonstrate formation of clusters.

Response: Please see response to comment 1.

9. The characteristics of MHCII cluster such as, cluster density, number of MHCII per cluster and fraction of MHCII participating in cluster formation process need to be determined. This has a direct link to the reported study related to cluster formation and physiological state of the cells.

Response: While this data was part of our cluster analysis, cluster statistics were indeed missing from the manuscript. We have now added new Figure S3c-e showing the 1) fraction of clustered localizations, 2) the cluster density per μm^2 and 3) the number of localizations per cluster, a proxy for the local accumulation of receptors.

10. Finally, I feel more work needs to be done for a complete study and the work better suits core virology journals.

Overall, the work lacks cluster analysis, determination of critical parameters related MHCII clustering, and ignores reported super-resolution study on Influenza A virus. In general, more work needs to be done for a complete conclusive study.

Response: We would like to thank the reviewer for providing constructive feedback, which let us perform the additional cluster analysis that now provides additional evidence and quantification of virus-receptor interaction, including extensive re-analyzation of our data. Receptor engagement of IAV that use a proteinaceous receptor is of great interest for a broad readership (Nat Microbiol. 2024 Oct;9(10):2626-2641; Cell Host Microbe. 2024 Jul 10;32(7):1089-1102.e10, Nature. 2019 Mar;567(7746):109-112., Nat Commun. 2024 May 27;15(1):4500.)

We would like to thank the reviewers for their feedback and provide a point-by-point response to the individual comments below.

Original Reviewers' comments are in black.

Our response is in blue, changes in the main manuscript and SI are highlighted in yellow.

REVIEWER COMMENTS

On the bright side, the cluster parameters are now provided. However, still a lot of other information is missing. See my comments below:

1. Fig. 4d-f is not time-lapse super-resolved reconstruction at varying time-points. Fig. 4d-f does not show the formation of clusters as claimed in the study. Moreover, Supplementary figure 4k-l is not the time-lapse image of the entire cell. Ideally, time-lapse imaging of an entire cell is likely to give visual evidence of cluster formation with time. This will be the conclusive proof.

Response: For the investigation of virus-receptor interaction, we primarily rely on live-cell PALM super-resolution microscopy, where the sparse photo-activation of fluorescent mEos proteins is used to localize and track receptors over a short period before mEos photobleaches (sptPALM). To allow precise localization, one prerequisite of sptPALM is the sparse activation density so that only one photoactivated protein should emit light within a diffraction-limited spot (diameter ~300 nm). The virus-binding site is such a diffraction-limited spot and hence the nature of the measurement is not compatible with the direct visualization of multiple proteins at once as they come together to form a cluster. To be able to visualize the dynamic developments at the virus-binding site, we thus reconstructed time-binned movies of our sptPALM data (shown in Supplementary Movies 1 - 4 and Supplementary Figure 4 k-n). Here, we combine many individual molecules, which allows the observation of cluster build-up as shown and quantified in Supplementary Figure 4 k-n.

Viruses are nanoscale entities and hence there is currently no evidence that virus-cell binding would have a global effect on the organization of receptors. Our PALM imaging was performed by focusing on the apical part of the cells where viruses are bound. Since only molecules in focus are detected, PALM single-molecule imaging of receptors on the entire cell at once is not possible. However, we do observe a larger area of the cell and have now analyzed the receptor clusters across the apical membrane. We did not find a difference in cluster size compared to cells not in contact with viruses (Rebuttal Figure 1). This confirms that viruses associate with larger clusters which is a very local effect occurring at the nanoscale within the virus-cell binding site.

Rebuttal Figure 1 Comparison of the mean MHCII cluster sizes determined on the apical membranes of uninfected and H18N11-infected cells. For the latter we show the mean cluster size at virus-bound surfaces as well as that across the apical membrane (global).

2. Supplementary figure 3 and 4 does not contain light transmission image. Moreover, the images quality is not good. Specifically, the representative widefield images of H18N11 viral particles (magenta) and MDCK-II cells are of poor quality. Drawing any conclusion from the imaging data is still doubtful.

Response: We apologize for not clarifying the missing light transmission images. The microscope we used does not have a brightfield modality. This was removed for steric and stability reasons. From the experience of the authors, also microscopes used in previous studies were not equipped with a brightfield modality, mostly for stability reasons (Sieben et al., PLoS Path, 2020 and Nat Meth, 2018). The health and shape of the cells is routinely checked using a brightfield microscope within the cell culture lab and before the samples are mounted for super-resolution microscopy. During the acquisition sessions, we acquire fluorescence widefield images before the PALM image stack to verify the position of the cell and the viruses. From these images, we extract only the positions of the virus particles, no further conclusions are drawn.

3. I still miss details related to the imaging system. For example, what is the power used for photoactivation and excitation of the mentioned lasers (405, 488, 561, 642nm). Moreover, information related to dwell-time, EMCCD gain, average number of molecules detected per frame and other details are missing.

Response: mEos3.2 was activated with 405 nm laser (power at objective back aperture of 0.1–5 μ W) and excited with 561 nm laser (power at objective back aperture of 0.1–2 mW) or with a 488 nm laser (power at objective back aperture of 0.1–2 mW). The sample was excited with a 642 nm laser (DiD) with a laser power at objective back aperture of 5–15 mW. The PALM image stacks were taken using an Andor iXon 897 EMCCD camera with an EM gain of 200 and an integration time of 30 ms/frame. We detected between 10-100 localizations per frame.

We have added the missing information to the manuscript.

4. Cluster identification in Supplementary Fig. 5 does not seem to be accurate. One can see lot more clusters than what is identified by the algorithm. Re-check chosen parameters in the cluster algorithm.

Response: The cluster identification is accurate, but we agree that the scatter plots as presented are not ideal for visualizing molecule clustering and may lead to misinterpretations. We have expanded Supplementary Fig. 5 and now include a plot of the MHCII localization density, which is exactly what is quantified by the clustering algorithm. A scatter plot, where each MHCII localization is color-coded according to its localization density, is more appropriate for the visual inspection of the data. In the new Supplementary Figure 5, we have added three examples of the algorithm identifying small and large clusters. The clusters are clearly visible in the density representation. There is no need to re-analyze the selected parameters in the clustering algorithm.

5. Still, the super-resolved images are very noisy.

Response: Super-resolved images from SMLM are reconstructed based on the position of the respective localizations. The localizations are filtered based on standard parameters such as the number of collected photons and the localization precision. The appearance of the super-resolved images depends on the nature of the molecule being investigated.